# Managing the Consequences of Oncological Major Surgery: A Short- and Medium-Term Skills Assessment Proposal for Patient and Caregiver through M.A.D.I.T. Methodology

**DOI:** 10.3390/bs12030077

**Published:** 2022-03-15

**Authors:** Gian Piero Turchi, Alessandro Fabbian, Rita Alfieri, Anna Da Roit, Salvatore Marano, Genny Mattara, Pierluigi Pilati, Carlo Castoro, Davide Bassi, Marta Silvia Dalla Riva, Luisa Orrù, Eleonora Pinto

**Affiliations:** 1Department of Philosophy, Sociology, Pedagogy and Applied Psychology, University of Padua, 35131 Padua, Italy; gianpiero.turchi@unipd.it (G.P.T.); fabbian.alessandro@gmail.com (A.F.); bassidavide94@gmail.com (D.B.); martasilvia.dallariva@unipd.it (M.S.D.R.); luisa.orru@unipd.it (L.O.); 2Unit of Surgical Oncology of the Esophagus and Digestive Tract, Veneto Institute of Oncology IOV—IRCCS, 35128 Padua, Italy; rita.alfieri@iov.veneto.it (R.A.); genny.mattara@iov.veneto.it (G.M.); pierluigi.pilati@iov.veneto.it (P.P.); 3Division of Upper Gastrointestinal Surgery, Department of Surgery, Humanitas Research Hospital, 20089 Milan, Italy; annadaroit@gmail.com (A.D.R.); salvatore.marano@humanitas.it (S.M.); carlo.castoro@humanitas.it (C.C.)

**Keywords:** upper GI–GI cancer, health, qualitative research, competences, text analysis, surgery, M.A.D.I.T. methodology

## Abstract

The effects of cancer surgery and treatment harm patients’ life and working ability: major causes of this can be intensified by the postoperative symptoms. This study, the first part of the HEAGIS project (Health and Employment after Gastrointestinal Surgery), proposes a method to assess patients and caregivers’ competences in dealing with postoperative course and the related needs to improve the adequate competences. In this observational study, an ad hoc structured interview was conducted with 47 patients and 15 caregivers between the third and fifteenth postoperative day. Oesophageal (38%), esophagogastric junction (13%), gastric (30%), colon (8%) and rectum (11%) cancer patients were considered. Computerized textual data analysis methodology was used to identify levels of competences. Text analysis highlighted three different levels (low, medium and high) of four specific types of patients and caregivers’ competences. In particular, the overall trend of the preview of future scenarios and use of resource competences was low. Less critical were situation evaluation and preview repercussion of own actions’ competences. Caregivers’ trends were similar. The Kruskal–Wallis test did not distinguish any differences in the level of competences related to the characteristics of the participants. Patients and caregivers are not accurate in planning the future after surgery, using personal beliefs rather than referring to physicians, and not recognizing adequate resources. The medium-low competences’ trend leads to unexpected critical situations, and patients could not deal with them in a maximally effective way. Both patients and caregivers should be taken over by healthcare professionals to improve patients’ competences and make the curative surgery effective in daily life.

## 1. Introduction

Despite the decrease in deaths caused, gastrointestinal (GI) and upper gastrointestinal neoplasms are the most commonly diagnosed tumours and carry a significant burden of symptoms [1,2]. These symptoms last even after curative treatments [3]: many studies underline the implications of surgery as elective treatment for GI and upper GI cancer on patients’ health experience [4,5,6]. Indeed, physical implications such as reduced tolerance to energy or activity, heartburn, diarrhoea, constipation, early feeling of fullness after eating and dumping syndrome are considered related to fatigue and worry, anxiety and depression, sleep disturbances and also difficulty reconstructing the experience after cancer diagnosis [7,8,9,10,11].

Moreover, studies show the relationship between these lasting symptoms after curative surgery and a poor health-related quality of life (HRQOL) [12,13,14]. Consequently, the effects of long-term treatment harm patients’ social lives and their ability to work. Effects have been observed on survivors’ employment activities, job productivity and personal finances after major surgery [15,16,17].

Furthermore, cancer survivors are about 1.4 times more unemployed than healthy participants and are the lowest percentage of patients resuming work within two years after diagnosis, which includes patients who have undergone cancer surgery [2,17,18,19,20]. The main causes of this unemployment can be job discrimination, difficulty in combining treatment with full-time work and physical or psychological limitations [15,17].

The previously reported data on cancer patients’ employment difficulties can be used as a starting point to deepen the social implications related to cancer surgery, starting from relationships with family and friends. In fact, relatives and caregivers of the patient are both involved in the effects of the intervention and in their management [9,21,22,23,24]: patients’ troubles can have consequences in all the daily relations and aspects of life that also involve other people (including eating, moving, daily activities, etc.). Relatives, friends, and caregivers live problematic situations together with the oncological patient, sharing feelings and difficulties that influence their way of showing support to the patient. For example, patients with greater social support from significant caregivers in family or between friends or mates or institutions/societies at baseline show more improvements in anxiety and depression management 12 months after surgery [21]. Furthermore, the literature shows family members suffer from emotional distress, with psychosocial disorders affecting at least 25% of the members [9], and that loneliness among caregivers of patients contributes to negative effects on their social, emotional and physical well-being [22]. Additionally, caregiver skills (in health, social, organizational, emotional and well-being relations areas) are often low or absent or they do not have adequate training to support patients at home [23]. Thus, caregivers are also involved in patients resuming HRQOL, and they should be considered as involved in oncological situations and their management.

Various types of psycho-oncologic interventions (individual psychotherapy, group and couple psychotherapy, relaxation training, psychoeducation, sole information) have a central role to help patients and caregivers in managing implications of surgery and consequent daily difficulties. Current literature suggests that these interventions can significantly improve HRQOL, emotional function (EF) and social function (SF) [21,24,25,26,27,28]. Several psychological approaches offer support to cancer patients; the general objective is to construct a patient’s self-representation without a pervasive stereotypical description of themselves focused on cancer disease. This is possible since quality of life (QoL) is “patients’ perception of their own position in life in the context of the cultural systems and the reference values in which they are inserted and in relation to their own goals, expectations, standards and interests” [29]. In other terms, the ability of patients to manage their new condition depends not only on the lasting symptoms but also on their ability to take control of the changes and learn how to manage the symptoms, rather than letting the symptoms limit every aspect of their lives. Referring, then, to what has been said before about caregivers, it is also possible to observe that the skills they deploy will have a key role in managing the situation (QoL) [9,21,22,23,24]. Hence, given this new complex life situation encountered after surgery, the needs of a care program arise, focused not only on the physical needs of patients but also on the mental, social and interactive dimensions that characterize this scenario [24]. Therefore, assessing the skills that patients and caregivers can use and express to face and manage the oncological diagnosis, deepening the interactive processes that lead to a healthy condition and a good QoL for the patient–caregiver dyad, is useful.

Providing a contribution in the management of this need, Padua University (Padua, Italy), Humanitas Research Hospital (Milan, Italy) and Veneto Institute of Oncology IOV —IRCSS (Padua, Italy) carried out a two-year research program, the HEAGIS project (Health and Employment after Gastrointestinal Surgery) [30]. HEAGIS aimed to offer a validated intervention model for the health management of the after-surgery oncological patient’s socioeconomic difficulties. The study we describe here, the beginning part of the main project, shows and comments on the results collected through a tool previously created and tested on a small sample of patients and caregivers [30] to assess the competences of patients and caregivers dealing with the postoperative course after surgery, specifically in patients with oesophageal, gastric and colorectal cancer. Moreover, the collected text data have been used for the second part of the main project, that is, the development of a closed-ended skills assessment questionnaire.

## 2. Materials and Methods

### 2.1. Methodological Background

The study is founded on dialogical science [31,32,33,34,35,36,37,38,39,40], a scientific approach that has its roots in the work of Harré and Gillet [41] and Berger and Luckmann [42], but most of all Wittgenstein [43] and Salvini [44]—thus, the interactionist scientific paradigm—and that studies language as the tool that generates sense of reality through interactions [33]. The methodology used, consistent with the paradigmatic reference, is M.A.D.I.T. [32,45,46,47,48,49], which allows researchers to analyse, describe and measure how the sense of reality [32] is generated by different interacting roles and voices involved in the oncological situation, i.e., oncological patients and caregivers [46,50,51]. Thus, M.A.D.I.T. allows researchers to study the ways with which the text links every single content to the other, and the consistency under the sense of reality is created. The modalities are the discursive repertories (RDs; 24; 32; 33; 44; 50) that can open the possibility of changing or maintain the sense of reality as itself, offering data (dialogical weight, dW; 32; 33; 39) about the generativity of the configurations of sense of reality that emerge from narrations.

Under this theoretical hat and considering the HEAGIS project aim, “health” is defined as the whole of the modalities, discursively intended, of configuring reality that consider, in terms of anticipation, the onset of disease and/or the generation of theories about disease [37]. This definition allows us to consider health not only as a body condition and/or disease but also the way in which the same condition is narrated among a reality of sense built through interaction. In the present study, the oncological diagnosis is a possibility that can emerge in a network of infinite events, which can undermine the reality of sense that has been maintained until the critical moment of diagnosis and that can be managed in medical terms, but especially interactive ones.

In this flow, the construct of *competence* is a dimension of health that enables the observation of how patients and caregiver, the main narrating voices in the oncological context analysed, build their way to manage oncological situation and future perspective, through the interactive modalities. Interactive modality is defined as a finite mode of construction of reality through the ordinary language, having a pragmatic value and valence of truth [32,39]. Starting from this theoretical definition, in the application of this definition to the research, these modalities acquire proper definition of discursive repertories. There is a definition for every discursive repertory (see Appendix A) [32,33].

Based on this study’s theoretical foundation, the use of language is seen creating those competences, namely technical–operational modalities and relationships acquired and developed in specific roles. Thus, patients and their caregivers are the roles using peculiar language in their roles and related peculiar actions put in place. In the surgical field, these modalities of language and consequently of actions by patients and caregivers have an impact on the postoperative management and the illness management. Therefore, configurations of reality are generated by interactive modalities, they concern the cancer management after surgical intervention and they are also composed by patients and caregiver competences in interacting with the events of cancer diagnosis, treatments and follow-up.

Hence, the research observation must consider a narrative focus which will deepen how the health and future perspective are described.

### 2.2. Measurements

Based on evidence from the literature on the state-of-the-art [52], coping styles and their adaptive value were considered. Thus, in dialogic science, coping skills, meaning those “thoughts and behaviours used to manage the internal and external demands” [53] in so-called stressful situations, were considered different narrations with consequent different actions. Focus on narration allowed us to identify four competences involved in the management of changes in health in the after-surgery (see Table 1 and Appendix A).

The combination of the use of these four competences leads patients to describe themselves not only in relation to the disease but also considering their activities, social and occupational features.

Thus, the above skills have been investigated within specific areas of investigation (see Table 2), extracted by current literature available about information needs in cancer patients [54].

The definition of these areas permitted the emergence of specific contents/arguments in the interaction with patient during the research.

Sixteen questions were constructed to describe the way patients and caregivers show their competence in managing the postoperative course [30]. Hence, the questions were given by a combination of the four skills and the four areas. Patients and caregivers had symmetric questions in the structured interview.

Competences were divided by degrees (three levels for each competence): every level (low, medium, high) is related to language frameworks representing the interactive ways used by respondents in order to manage their health [30]. Discursive repertories, indeed, are divided into classes: maintenance, hybrid and generative. The former are modes of language use that generate, in discourse, a stable and immutable reality of sense (contributing, for example, to the generation of a “cancer patient” reality of sense in every aspect of a person’s life, which will consequently operate under this assumption), while the latter account for interactive modes that promote a change from the current state of things (contributing, for example, to the generation of a reality in which the individual can tell his story as a “cancer patient” but also as a family man, a husband who helps around the house, a friend with whom to share a walk). Hybrid repertories can have both maintenance and generative valence, depending on the repertories they link to in their use.

Starting from the analysis of the use of language of the respondents, it has been possible to define criteria that would allow to identify different levels of expression of the skills described above, i.e., high, medium and low competence, depending on the use of language made by the respondent. Generative modalities, indeed, allow us to anticipate a more effective management of the postoperative period than others, where the effectiveness is given both by compliance with the treatment and by the possibility to tell the story of the individual, not only as a “cancer patient”. Vice versa, low levels of competence match with usage of language modalities that allow us to anticipate the maintenance of a sense of reality where the role of “cancer patient” becomes pervasive in every aspect of life and a minimal degree of effectiveness in managing one’s own life after diagnosis and surgery. As said, the use of hybrid modalities will allow us to anticipate different scenarios depending on the repertories they will link with [30]. For further details and examples on competences’ stratification criteria, see Appendix A and the paragraph 4.

### 2.3. Study Design and Participants 

The present cross-sectional study was conducted between April and July 2019. Patients were recruited during their hospital stay after the surgery. In this study, a convenience sample was considered: consecutive 47 eligible patients and 15 caregivers were interviewed. This difference was due to the need for simultaneous administration of the questionnaires for patients and caregivers. There were no exclusion criteria for caregivers. The inclusion criteria of the patients were: 18 years or older; Italian language comprehension; oesophageal, gastric or colorectal cancer diagnosis; eligible for curative surgery; metastases free; hospitalized between the third and fifteenth postoperative day.

The exclusion criteria were: younger than 18 years old; lack of Italian language comprehension; other cancer diagnosis (not oesophageal, gastric, or colorectal cancer diagnosis); eligible for palliative surgery; metastases presence; hospitalized before the 3rd and after the 15th postoperative day. Patients were presented with the sixteen questions in the form of a structured interview, which was conducted and transcribed verbatim by a trained psychologist. The interviewer was prepared in order to reformulate questions without overwriting the questions of the questionnaire, maintaining the adherence to the text, not only when transcribing answers but also when asking questions, without addressing answers. Patients and caregivers were interviewed separately, at a different time and place. Each interview was about 15 min long. The module for the expression of informed consent was presented to and subscribed by all participants prior to the study.

### 2.4. Data Analysis

The text generated by the questions has been analysed using M.A.D.I.T. methodology [33,45,50], which, as said before, allows one to observe and describe the language use modalities adopted by respondents to build a reality of sense concerning the oncological situation management and its consequences. More specifically, through this methodology and the semiradial table of the discursive repertories, it has been possible to collect and describe the discursive repertories that characterize the narrations of patients and caregivers about the skills they express in managing their health condition after surgery, as previously mentioned: the broader the use of discursive modalities that vary from maintaining reality to generating different possible realities, the higher the degree of expression of the single competence will be. Once the repertories [50] have been identified, it has been possible to assess the level of competence of the respondents using the criteria mentioned above (see Appendix A).

After competences levels were identified and made measurable, through text analysis as described above, the Kruskal–Wallis [50] test was used in order to verify the competences distribution in relation to patients’ demographic and clinical conditions.

## 3. Results

### 3.1. Patients’ Competences

Patients’ profiles of competences have been described in Table 3 and patients’ characteristics completed of all variables considered in Appendix A. The Mann–Whitney test and Kruskal–Wallis test were used—respectively for two data samples and for more than two data samples—and did not show any competence distribution in relation to patients sex, age, level of education, employment, neoadjuvant therapy, cancer localization and type of surgery. Other variables were considered. Marital status and having children were analysed to detect the presence of resources in the nuclear family. The presence of comorbidities for each patient at the hospital admission for surgery was analysed and found not significant. In addition, peculiar characteristics of comorbidities were considered: comorbidities with a poor prognosis (other neoplasms and cardiovascular comorbidities), comorbidities implying a peculiar disease management by patient (chronic comorbidities and past surgery) and psychiatric comorbidities. The analysis did not distinguish any differences in the level of competences related to patients’ comorbidities.

The overall trend of the competences’ expression is low, reaching its peak in the preview of future scenarios competence (see Table 3). In contrast, considering the preview repercussions of own actions, it is less critical if compared to the former: patients seem to be more accurate when they have to evaluate the relapses of their choices in daily life, rather than future scenarios linked to their conditions. Last, the use of resources has been less critical in the various areas: patients may therefore be able to recognize what resources could be used to deal with a peculiar situation. The situation evaluation is very much linked to other competences since it encompasses those modalities necessary for observation, description and analysis of a situation, that are steps also required for effective use of use of resource, preview of future scenarios and preview repercussion of the own actions. Therefore, the situation evaluation is very much linked to other competences: the level of this competence is low, and this has an impact on other competences, since patients describe their condition using opinions and judgement rather than referring to what was said by physicians.

### 3.2. Caregivers’ Competences

Considering the caregivers group, the trend of the level of competence (see Table 4) is very similar to the patient’s one, and this has been verified thorough the application of the Kruskal–Wallis test.

As for the patients, the most critical caregivers’ competence is the preview of future scenarios, while the less critical is the use of resources. Situation evaluation also has a low level, and this affects the other competences since it is related to the skills of anticipation and use of resources. The preview repercussion of own actions turns out to be less critical compared to the preview of future scenarios: caregivers also tend to be more precise in imagining what could happen if placed in a condition where their choice is required.

## 4. Discussion

As described above, the most critical competence was preview of future scenarios. When patients do not anticipate potential future developments of a state such as possible critical aspects arising after the intervention, they will not be able to identify strategies to face the difficulties. In this way, patients must find solutions with urgency, with a different effectiveness compared to strategies developed and reasoned without urgency. Not anticipating difficulties leads patients to face them only once they appear, without having the knowledge to deal with them, going by trial and error, possibly resulting in inadequate behaviour.

Unlike other studies in other fields, in these results there are no differences between men and women and between younger and older patients about concerns about the consequences of a procedure [55].

The lack of anticipation is also represented in the results about the preview repercussions of one’s own actions, that even if it is less critical, it is still on the low level. The actions derived from this competence are attributable to preventive health behaviour in the current literature [56]: the results of this study show that the previews of patients with GI and upper GI cancers could lead to situations not seen or also emergency situations due to inadequate actions for postoperative conditions. Hence, patients could not deal with these situations in a maximally effective way, since they are unforeseen consequences of decisions taken without considering implications or considering only their own opinion and beliefs.

This is related to the situation evaluation competence, which has an impact on all other competences, since its low levels mean that patients describe their condition using opinions and judgment rather than referring to what the physicians and other healthcare professional roles said or considering desires or habits more than the consequences of surgery. In this way, patients’ erroneous beliefs on the postoperative course may lead to a misleading communication with the surgeon. Thus, using RDs as judgement or opinion (so, expressing a low competence level) could create a difficulty of patients, caregivers and the health professional adopting a shared use of management elements related to the same plan.

The low level of situation evaluation competence and the preview of future scenarios is consistent with the literature stressing that monitoring and self-monitoring precede the evaluation of progress towards goals and self-reinforcement of the progress made [57], so the inaccurate situation evaluation could also lead to blind preview of scenarios.

The low level of the competence preview of future scenarios also has an impact on the possible effectiveness on the use of resources: despite this, competence is the less critical issue, since there is little anticipation of difficulties, and patients will activate external resources only in the urgent need of such, i.e., limiting the support that can be given by the physicians in the postoperative course. In this regard, the surgeon could be asked too late for consultation by the patient or could be not asked at all. The less critical use of the resources competence is confirmed by studies in which availability of resources can be triggered, even in critical situations [58,59].

These implications for the low level of competences are also transferable on the caregivers’ group, since the results were similar. A low level for preview of future scenarios, such as for preview repercussions of one’s own actions can lead to inadequate support since difficulties are faced only when they happen, in an emergency situation. Therefore, as seen above, the use of resources that caregivers have access to will be less effective, and the evaluation of the situation based on personal opinions and beliefs can lead to inappropriate suggestions to patients.

In addition, the low levels of competences by both the caregivers and the patients can correspond to a maintenance of the patient’s choices and their narration in terms of “sickness”. This can lead to an abandonment of the patient’s own activities since they are considered by both the patient and the family as not proper for a “sick person”. When the surgery is curative, patients and their caregivers focus only on the surgical intervention, as it is the event generating health, not considering implications during following months. In this way, they overlap health with the results of surgery and delegate both the management of the consequences of surgery and the daily life activities recovery to health professionals, as the medium level of competence of use of resources suggests.

This modality is not enough for patients’ activities recovery when preview of future scenarios and preview repercussions of their own actions are low: this entails lack of outlining the consequences in the future of actual choices. Thus, patients and caregivers do not ask questions about the present nor the future situation, and they do not identify doubts on the different possibilities of surgical consequences management.

Furthermore, since results showed a similar trend of competences between patients and caregivers, it is possible to assume that the support the patient will receive at home will also focus on avoiding certain activities, without actually knowing if it is appropriate, using only hopes and opinions. Instead, caregivers could encourage patients to try to do certain activities, disregarding actual consequences that they have not anticipated.

In general, findings from this study suggest considering patients’ risk perception a central construct in many current health behaviour theories and support the suggestion about poorer outcomes in patients who are more likely not to use the competences to preview and manage possible postoperative risks. Thus, through the sixteen questions about the four competences in clinical, daily activities, family and work areas, we were able to identify what in literature is called cognitive and affective risk perception [60]. For upper GI and GI cancer patients, low levels of competences elucidate the poor HRQOL evaluated by patients with lasting effects evidenced by literature and stress how patients’ information and support needs are significant and continuous in the postoperative [61]. From all competences mentioned above is derived the possibility that the HRQOL of the patients will be limited by the way they and their relatives describe the situation, focusing on the condition of sickness and not on what to do to generate new ways of narration about their condition. Other works highlight the need of psychological intervention in patients with low coping strategies, in particular for cancer patients with poor cancer prognosis [62]. Therefore, supporting patients and caregivers in the postoperative and after hospital dismission can improve the short-term postoperative HRQOL and the patient–physician communication in order to prevent cancer recurrence and manage the symptoms in a more effective way. Indeed, improving patients’ and caregivers’ competences in the clinical area could overwhelm communication barriers, allowing the professionals to anticipate the future possible scenarios of patients in the management of the surgery results.

This study addresses an area that should be improved in psycho-oncology and in psychology applied to patients who have undergone oncological surgery. This area concerns patients’ and caregivers’ skills in reflecting psychological processes involved in daily life management after major surgery for cancer. In particular, this is the first study underlining the role of patients’ skills on management after major surgery for GI and upper GI cancer.

Not only patients’ but also caregivers’ skills were considered, and questions of the interview were symmetrical for patients and caregivers and administered at the same time. Indeed, the methodology used allowed us to highlight patients’ and caregivers’ approach to disease beyond their general judgment about the neoplasm and treatment, stressing the critical aspects in their approaches, which could potentially invalidate surgical results.

Findings given by this study are limited by the small sample size for each cancer localization (oesophageal, gastric and colorectal). In addition, the number of caregivers is reduced compared to the sample number of patients, and both sample numbers, overall, are reduced. A second limitation is that the work area was investigated only if the patient was regularly taken or VAT registered during administration. Out of 47 total patients, 29 were nonworkers, so 18 patients were asked about the work area. Therefore, the number of patients should be improved for each cancer localization and working status and made equivalent.

## 5. Conclusions

In conclusion, this study highlighted a lack, both in patients and caregivers, in the competences to deal efficiently with after-surgery implications. Implementing psychological support which works on emotional impacts could be useful, but it is necessary to also offer a more specific intervention which can help patients in finding better strategies to overcome new difficulties they will be facing once returned to their home. Therefore, future directions in clinical activity should also consider the competences described in the support given to the patients, since they can promote both a clinical recovery and a return to social and work activities.

The study uses a questionnaire requiring an in-depth interview. Since project HEAGIS aims to create a replicable and validated tool for evaluation of patients’ competences to deal with postoperative course (transversely usable by health professionals), the texts gathered in this study will be used to build the answers of a closed-ended questionnaire evaluating the competences of patients in a quicker and more efficient way. Different options in different answers will reflect the different levels of competences, giving a graphic and descriptive output for health professionals.

This tool will be applied in a longitudinal observational study, conducted with patients from surgery to 9 months after surgery, and in an interventional study, where it will be used to measure the results of a psychological support intervention.

## Figures and Tables

**Table 1 behavsci-12-00077-t001:** Management Health Competences.

Preview of future scenarios	how patient depicts the development of the present situation
Situation evaluation:	how patient describes his/her situation and evaluates what to do
Preview repercussion of their own actions:	how patient depicts implications of his/her actions regarding his/her condition
Use of resources	how patient considers the resources on which he/she can rely (i.e., family, doctors, etc.) as a support to change critical issues in his/her condition

**Table 2 behavsci-12-00077-t002:** Investigation areas.

Clinical Area	Physiological, pathological and hospital procedures aspects involved in GI and upper GI neoplasms surgery, for example, symptoms, procedures, hospital access, etc.
Daily Activities Area	The activities carried out by patient in his/her own life, for example, passions, social encounters, intellectual or physical activities, etc.
Family Area	The interactions within the family, evaluated in response to surgery for neoplasm
Work ^1^ Area	The aspects regarding the working position: working environment, tasks performed, working hours, etc.

^1^ This area has been investigated only if the respondent at the time of the interview was regularly employed.

**Table 3 behavsci-12-00077-t003:** General trend of competences expression among the sample.

Competences	Low	Medium	High
Preview of future scenarios	76%	22%	2%
Situation evaluation	57%	26%	17%
Preview repercussions of the own actions	60%	26%	14%
Use of resources	50%	29%	21%

**Table 4 behavsci-12-00077-t004:** General trend of competences among the caregivers.

Competences	Low	Medium	High
Preview of future scenarios	84%	16%	0%
Situation evaluation	51%	33%	16%
Preview repercussion of the own actions	55%	33%	12%
Use of resources	47%	45%	8%

## Data Availability

Data are available with an explicit and formal request to the corresponding author, due to privacy policy.

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
