# Peer review of "Managing the Consequences of Oncological Major Surgery: A Short- and Medium-Term Skills Assessment Proposal for Patient and Caregiver through M.A.D.I.T. Methodology"

_behavsci, 2022, doi:10.3390/bs12030077_

Round 1
Reviewer 1 Report
The paper presented is interesting and well-founded. The tables and supplementary material are clarifying and help understanding. Moreover, its applicability for the improvement of patients' quality of life is clear.
Some considerations are indicated:
In the section "Patients´ competences" (lines 280 and 281) it is stated "The situation evaluation is very much linked to other competences:" More specificity about which competences it refers to is requested.
It is recommended to provide more information on patient profiles and deepen its relationship with discursive repertories. For example, if they have a partner and children, presence of other diseases or people close to them with cancer. Similarly, it would be useful to know in more detail the profile of caregivers.
A greater description of the context of the interview and the interviewer would be of interest for a better contextualization
Author Response
The paper presented is interesting and well-founded. The tables and supplementary material are clarifying and help understanding. Moreover, its applicability for the improvement of patients' quality of life is clear.
Some considerations are indicated:
1) In the section "Patients´ competences" (lines 280 and 281) it is stated "The situation evaluation is very much linked to other competences:" More specificity about which competences it refers to is requested.
We thank the reviewer for this recommendation. Within the theoretical frame adopted, the “situation evaluation” is described as the competence which implies an observation of the events, a description of them and a consequent analysis in a way that other roles (i.e. caregivers, health care professionals) can understand and share. For this reason, the “situation evaluation” encompasses those steps, prodromal for the other competencies described in the study (use of resource, preview of future scenarios, preview repercussion of the own actions), relying on the situation evaluation. At the same time, situation evaluation and the other three competencies can be considered apart from each other, since they do not exhaust completely one each other.
In accordance to the reviewer’s suggestion, we implemented this specification in the text (page 7):
“The situation evaluation is very much linked to other competences since it encompasses those modalities necessary for observation, description and analysis of a situation, that are steps also required for effective use of resource, preview of future scenarios and preview repercussion of the own actions. Therefore, the level of this competence is low, and this has an impact on other competences, since patients describe their condition using opinions and judgement rather than referring to what was said by physicians”.
2) It is recommended to provide more information on patient profiles and deepen its relationship with discursive repertories. For example, if they have a partner and children, presence of other diseases or people close to them with cancer. Similarly, it would be useful to know in more detail the profile of caregivers.
According to this recommendation, we implemented further information in patient profiles. We added these descriptive variables in analysis of potential correlation between patients’s profile and level of competences, as already done for sex, age, employment, neoadjuvant therapy, cancer localization and type of surgery, variables resulting not correlated to the levels of competences. This is now specified on page 7: “Patients' profiles of competencies have been described in Table 3 and patients characteristics completed of all variables considered in Supplementary Materials S2. The Mann-Whitney test and Kruskal-Wallis test were used - respectively for two data samples and for more than two data samples - did not show any competence distribution in relation to patients sex, age, employment, neoadjuvant therapy, cancer localization and type of surgery. Other variables were considered, as marital status and having children, to detect the presence of resources in the nuclear family. The presence of comorbidities for each patient at the hospital admission for surgery was analysed and found not significant. Also peculiar characteristics of comorbidities were considered: comorbidities with a poor prognosis (other neoplasms and cardiovascular comorbidities), comorbidities implying a peculiar disease management by patient (chronic comorbidities and past surgery) and psychiatric comorbidities. The analysis did not distinguish any differences in the level of competences related to patients’ comorbidities.”
3) A greater description of the context of the interview and the interviewer would be of interest for a better contextualization.
Consequently to this proper suggestion, we enforced this description in Materials and Methods (page 6, paragraph 2.3 Study design and participants):
“Patients were presented with the sixteen questions in the form of a structured interview, which was conducted and transcribed verbatim by a trained psychologist. The interviewer was prepared in order to reformulate questions without overwriting the questions of the questionnaire, maintaining the adherence to the text, not only when transcribing answers but also when asking questions, without addressing answers. Patients and caregivers were interviewed separately, in different time and place”.
Reviewer 2 Report
A decently performed study on patients affected by gastrointestinal cancers and their caregivers. Overall if found this method to assess patients and caregivers’ competencies in dealing with the postoperative course and the related needs very useful. The main limitation of the paper is the low number of participants, but this query is not addressable.
I suggest accepting the paper
Author Response
We thank the reviewer for this evaluation.
Reviewer 3 Report
Innovative and interesting study on the postoperative skills levels of cancer patients and caregivers. However, I think the study could be improved with the following suggestions:
1) The title does not seem adequate to the study. I think that the authors identified levels of competence of cancer patients and caregivers in the short and medium term, so the title should reflect this data. Reformulating the title will be important for the impact of the article;
2) the text in the abstract on line 25 is not aligned;
3) Line 115 until line 141 there is text duplication;
4) I suggest that in item 2.2 Measurements, that they synthesize the text. It seems like a lot of explanation that could be more succinct, making the reading easier;
5) In line 247 the authors talk about interviews with 47 patients and 15 carers. There is a detail that should be mentioned for transparency in data reporting: who interviewed the patients and families? Did you have any preparation to avoid biases?
6) In line 319 and following, the authors refer to the low levels of competence of the patients in describing their health condition, basing it more on their opinions and beliefs than on what the doctor tells them. Can you find any reason for this? Why only what the doctor says? Isn't there information from other health professionals that influence this data?
7) In line 332 the authors refer to the doctor as a resource in the postoperative period. Also here, were there no other resources, other health professionals that serve in the future scenario for patients and caregivers? We are talking about the current architecture of health care delivery and this is multi professional, and in this scope the resources are multiple, not centred only on the doctor.
8) Lines 393 to 397 are ideas that have been repeated before.
9) The fact that you include patients with oncological pathology from the oesophagus to the rectum, do you not think that there may be some bias with regard to the competencies of the patient and caregivers, i.e., do you not think that there may be differences between a patient with oesophageal oncological pathology who has undergone surgery and a patient with colon oncology who has been left with a colostomy?
10) In line 410 the authors mention that the interview was carried out by psychologists, something that I think should be written in point 2.1 as I question in point 5) of this document;
11) Finally, in the conclusions, it seems to me that the last paragraph should be the first and the rest of the text, which refers to future studies, should follow.
Author Response
Innovative and interesting study on the postoperative skills levels of cancer patients and caregivers. However, I think the study could be improved with the following suggestions:
1) The title does not seem adequate to the study. I think that the authors identified levels of competence of cancer patients and caregivers in the short and medium term, so the title should reflect this data. Reformulating the title will be important for the impact of the article;
Thanks for the proper recommendation, the short and medium term reference has been added in the title.
2) the text in the abstract on line 25 is not aligned;
This reported typo has been modified.
3) Line 115 until line 141 there is text duplication;
Text has been changed and the typo corrected.
4) I suggest that in item 2.2 Measurements, that they synthesize the text. It seems like a lot of explanation that could be more succinct, making the reading easier;
Considering this proper suggestion, we cut some parts maintaining those information concerning the measurements plan, useful to make the study replication possible.
5) In line 247 the authors talk about interviews with 47 patients and 15 carers. There is a detail that should be mentioned for transparency in data reporting: who interviewed the patients and families? Did you have any preparation to avoid biases?
We thank for the suggestion given, enforcing the Materials and Methods section (page 6, paragraph 2.3 Study design and participants, from line 247). The text was so implemented:
“Patients were presented with the sixteen questions in the form of a structured interview, which was conducted and transcribed verbatim by a trained psychologist. The interviewer was prepared in order to reformulate questions without overwriting the questions of the questionnaire, maintaining the adherence to the text, not only when transcribing answers but also when asking questions, without addressing answers. Patients and caregivers were interviewed separately, in different time and place”.
6) In line 319 and following, the authors refer to the low levels of competence of the patients in describing their health condition, basing it more on their opinions and beliefs than on what the doctor tells them. Can you find any reason for this? Why only what the doctor says? Isn't there information from other health professionals that influence this data?
Based on the theoretical background, modalities used by patients, therefore the level of competences, give reason of how patients consider what is said: when peculiar discursive repertoires are used, for example, the only matter considered by patients are their own beliefs and thoughts, while when other discursive repertoires are used, information given by health professionals are considered in a shared way and not distorted by patients.
Considering the reviewer hints, the text (pages 8 and 9) has been changed to make these aspects stressed by reviewer, clear.
7) In line 332 the authors refer to the doctor as a resource in the postoperative period. Also here, were there no other resources, other health professionals that serve in the future scenario for patients and caregivers? We are talking about the current architecture of health care delivery and this is multi professional, and in this scope the resources are multiple, not centred only on the doctor.
We thank the reviewer for this consideration: in the discussion we cited the physician but the questions in the questionnaire and the comment encompasses all health professionals serving in the postoperative scenario, as underlined by reviewer. We stressed this aspect also in the text (page 8)
8) Lines 393 to 397 are ideas that have been repeated before.
Thanks for this warning, we eliminated this part and modified the text (page 10).
9) The fact that you include patients with oncological pathology from the oesophagus to the rectum, do you not think that there may be some bias with regard to the competencies of the patient and caregivers, i.e., do you not think that there may be differences between a patient with oesophageal oncological pathology who has undergone surgery and a patient with colon oncology who has been left with a colostomy?
Considering how competences are defined by Discursive Repertories, i.e. language modalities through which every human being build the reality, and that they are 24 (and not endless), there are no differences between patients with different treatment or cancer type, because they all use these 24 modalities in order to build the cancer reality they live. As the reviewer said, diversity is manifested in the contents that patients use to create reality through those 24 repertoires: this is the reason why we collected text in this project phase, before building the closed ended questionnaire using their same contents, and then added a support intervention (papers about these phases are in revision).
Moreover, in the proper example done in the reviewer question, the digestive tract neoplasms considered in the study all consider different devices patients should manage: a jejunostomy after surgery for oesophageal and gastric tumours or a definitive or not definitive colostomy are possible. Therefore, anyway a patient with a digestive tract neoplasm has to face with definitive or temporary devices related to digestive tract.
10) In line 410 the authors mention that the interview was carried out by psychologists, something that I think should be written in point 2.1 as I question in point 5) of this document;
Please see the answer to comment 5.
11) Finally, in the conclusions, it seems to me that the last paragraph should be the first and the rest of the text, which refers to future studies, should follow.
We thank the reviewer for the recommendation: we moved the paragraph as suggested.